# Using viral sequence diversity to estimate time of HIV infection in infants

**Magdalena L. Russell**[1,2], **Carolyn S. Fish**[3], **Sara Drescher**[4,5], **Noah A. J. Cassidy**[3], **Pritha Chanana**[6], **Sarah Benki-Nugent**[7], **Jennifer Slyker**[7,8], **Dorothy Mbori-Ngacha**[9], **Rose Bosire**[10], **Barbra Richardson**[7,11,12], **Dalton Wamalwa**[7,9], **Elizabeth Maleche-Obimbo**[9], **Julie Overbaugh**[3], **Grace John-Stewart**[7,8,13,14], **Frederick A. Matsen, IV**[1,5,15,16‡*], **Dara A. Lehman**[3,7‡*]

1 Computational Biology Program, Fred Hutch Cancer Center, Seattle, Washington, United States of America, 2 Molecular and Cellular Biology Program, University of Washington, Seattle, Washington, United States of America, 3 Division of Human Biology, Fred Hutchinson Cancer Center, Seattle, Washington, United States of America, 4 University of Washington Medical Center, Seattle, Washington, United States of America, 5 Howard Hughes Medical Institute, Seattle, Washington, United States of America, 6 Bioinformatics Shared Resource, Fred Hutch Cancer Center, Seattle, Washington, United States of America, 7 Department of Global Health, University of Washington, Seattle, Washington, United States of America, 8 Department of Epidemiology, University of Washington, Seattle, Washington, United States of America, 9 Department of Pediatrics and Child Health, University of Nairobi, Nairobi, Kenya, 10 Centre for Clinical Research, Kenya Medical Research Institute, Nairobi, Kenya, 11 Department of Biostatistics, University of Washington, Seattle, Washington, United States of America, 12 Vaccine and Infectious Disease Division, Fred Hutch Cancer Center, Seattle, Washington, United States of America, 13 Department of Pediatrics, University of Washington, Seattle, Washington, United States of America, 14 Department of Medicine, University of Washington, Seattle, Washington, United States of America, 15 Department of Genome Sciences, University of Washington, Seattle, Washington, United States of America, 16 Department of Statistics, University of Washington, Seattle, Washington, United States of America

‡ These authors are co-senior authors on this work.
* matsen@fredhutch.org (FAM); dlehman@fredhutch.org (DAL)

**Data Availability Statement:** All modeling and analysis code implemented in this study are available on GitHub at https://github.com/matsengrp/infection-timing. The rstan available at

## Abstract

Age at HIV acquisition may influence viral pathogenesis in infants, and yet infection timing (i.e. date of infection) is not always known. Adult studies have estimated infection timing using rates of HIV RNA diversification, however, it is unknown whether adult-trained models can provide accurate predictions when used for infants due to possible differences in viral dynamics. While rates of viral diversification have been well defined for adults, there are limited data characterizing these dynamics for infants. Here, we performed Illumina sequencing of *gag* and *pol* using longitudinal plasma samples from 22 Kenyan infants with well-characterized infection timing. We used these data to characterize viral diversity changes over time by designing an infant-trained Bayesian hierarchical regression model that predicts time since infection using viral diversity. We show that diversity accumulates with time for most infants (median rate within *pol* = 0.00079 diversity/month), and diversity accumulates much faster than in adults (compare previously-reported adult rate within *pol* = 0.00024 diversity/month [1]). We find that the infant rate of viral diversification varies by individual, gene region, and relative timing of infection, but not by set-point viral load or rate of CD4+ T cell decline. We compare the predictive performance of this infant-trained Bayesian hierarchical regression model with simple linear regression models trained using the same infant data, as well as existing adult-trained models [1]. Using an independent dataset from an additional

https://mc-stan.org/ and rethinking R packages at doi: 10.1201/9781315372495 were particularly helpful. Filtered sequencing reads have been deposited into the National Institutes of Health Sequence Read Archive under accession code PRJNA1032485.

**Funding:** This work was supported by grants from the National Institutes of Health: R01 HD094718 for GJS and DAL, R01 AI076105 for JO, R01 AI146028 for MLR and FAM, P30 CA015704 to the Genomics and Bioinformatics Shared Resource (RRID:SCR_022606) of the Fred Hutch/University of Washington Cancer Consortium, and S10OD028685 to Fred Hutch scientific computing. FAM is an Investigator of the Howard Hughes Medical Institute (HHMI). The HHMI Medical Research Fellows Program contributed support for SD. The funders had no role in the study design, data collection and analysis, decision to publish, or preparation of the manuscript.

**Competing interests:** I have read the journal's policy and the authors of this manuscript have the following competing interests: FAM is an Investigator of the Howard Hughes Medical Institute (HHMI).

15 infants with frequent HIV testing to define infection timing, we demonstrate that infant-trained models more accurately estimate time since infection than existing adult-trained models. This work will be useful for timing HIV acquisition for infants with unknown infection timing and for refining our understanding of how viral diversity accumulates in infants, both of which may have broad implications for the future development of infant-specific therapeutic and preventive interventions.

## Author summary

Knowledge of the timing of HIV infection is crucial for improving our understanding of viral transmission and pathogenesis, especially in infants. In this group, viral load levels have been found to be much higher than in adults and vary based on age and mode of infection. In this study, we explore viral diversity dynamics during the early stages of pediatric HIV infection. Inspired by previous studies in adults, we develop infant-specific models that measure rates of viral diversification and use these inferred rates to estimate infection timing. Applying these models to a cohort of Kenyan infants, we successfully estimate their infection timing more accurately than existing adult-specific models. We also show that viral diversity accumulates much faster in infants compared to adults. This work provides new insights into how the HIV sequence diversifies in infants, offering valuable information for understanding differences in viral pathogenesis, transmission, and disease progression between infants and adults. These findings also highlight the importance of considering these differences when developing methodologies for future studies related to HIV infection timing across different age groups, as failing to do so may result in incorrect conclusions regarding the timing of pediatric infections.

## Introduction

Infants born to mothers living with HIV are at risk for acquiring HIV in-utero or during delivery and breastfeeding. Previous work has shown that viral load and pathogenesis vary based on age and mode of infection [2–4]. Specifically, infants infected prior to two months of age have substantially higher HIV RNA levels compared to later-infected infants and adults [2–4]. However, outside of these closely monitored cohorts, most newly-diagnosed infants with HIV have had an established HIV infection for an unknown duration. Defining the time of infection typically requires regular infant testing which is not always implemented. Knowledge of the timing of HIV acquisition is important for improving our understanding of viral transmission and pathogenesis in infants.

Adult studies have established that HIV sequence diversity increases over time [1,5–18]. Infections typically start with infection of just one or a few variants due to a bottleneck during HIV transmission that results in a relatively homogeneous viral population during the early stages of acute HIV infection [19–22], a concept that was first described for infants [23–27]. The error-prone nature of the reverse transcriptase enzyme and pressure from the host immune response causes the viral population to diversify rapidly over time [28]. During early infection, viral diversity has been shown to increase in an approximately linear fashion [6]. This trend has been observed using various sequencing platforms and measures of sequence diversity including the fraction of polymorphic nucleotides [1,11,13,14,16,18], high-resolution melting [15], and average pairwise diversity [1,10,12,17]. Of these measures, average pairwise

diversity, which is the probability that any two sequences will have different nucleotides at a specified position, averaged over all positions in the sequence, is arguably the most interpretable and easy to calculate. In fact, adult studies have shown that linear regression models using average pairwise diversity can be used to accurately estimate infection timing [1,18].

In contrast to adults, rates of viral diversification in infants are not well understood, despite extensive efforts to characterize other differences between infant and adult HIV infections. For example, HIV set-point viral load levels have been found to be substantially higher for infants compared to adults [2,3]. Because infants are infected in the presence of maternal antibodies, viruses may escape immune pressure prior to infection [27] and, thus, possess relatively higher replicative fitness [29,30]. Further, compared to acutely-infected adults, infants are slower to suppress virus [31] which may be related to the fact that infants possess a developing immune system at the time of infection. Because of these many differences, rates of viral diversification in infants likely differ from adults [32]. It is unknown whether viral diversity in infants increases in an approximately linear fashion during early infection and whether it can be used to estimate pediatric infection timing.

In this paper, we investigate how viral average pairwise diversity (APD) changes with time in a cohort of Kenyan infants, and explore whether rates of viral diversification can be used to accurately estimate infection timing. We designed a Bayesian hierarchical regression model to measure rates of viral diversification and trained this model using sequence data collected at 2–3 timepoints post HIV acquisition from 22 infants with well characterized timing of HIV infection. With this infant-trained model, we show that sequence diversity accumulates with time for most infants and that the rate of this accumulation varies by individual, gene-region, and mode of infection, but not by set-point viral load or rate of CD4+ T cell decline. We demonstrate that this model has well-calibrated uncertainty estimates, meaning that it delivers accurate confidence intervals. Further, we compared the ability of this infant-trained Bayesian hierarchical regression model to estimate pediatric infection timing with simple linear regression models trained using the same infant data, as well as existing adult-trained models [1]. We show that both infant-trained model types can more accurately estimate infant time since infection than the existing adult-trained models, although both have non-trivial levels of prediction error.

## Results

### Data overview

Kenyan children born to mothers living with HIV that were enrolled in cohort studies between 1992–2002 (see Methods) were followed from birth through age two with frequent HIV testing and banked plasma sampling [33,34]. Because antiretroviral therapy (ART) was not yet standard of care in Kenya during the time frame, the children were all ART-naive for the duration of monitoring, however, several of their mothers received a short course of ART during pregnancy. Frequent HIV testing (every 6–12 weeks) allowed us to estimate the time of HIV infection as the midpoint between the last negative test and the first positive test. We take this estimated time of HIV infection as the "true" infection time. For each infant, we used Illumina sequencing to calculate average pairwise viral diversity measures within 3 different regions across *gag* and *pol* from plasma HIV RNA (see Methods).

The model training data set consisted of sequences from 22 infants that had plasma samples available from 2–3 timepoints following HIV acquisition. Of these 22 infants, 11 were infected in-utero (log10 set-point viral load ranging from 5.72–6.52) and 11 were infected postpartum (log10 set-point viral load ranging from 5.29–7.80). The model testing data set consisted of

sequence data from only a single time point following HIV acquisition from 15 additional infants, of which 2 and 13 were infected in-utero or postpartum, respectively.

## Overview of modeling approach to quantify rates of viral sequence diversification in infants

As described above, the training data set contains viral sequence diversity (APD) measures sampled from multiple time points and gene regions for each individual (Fig 1A). Because of the multi-dimensional nature of these data, we chose to quantify rates of viral diversification using a Bayesian hierarchical model which affords us the flexibility of measuring the relationship between viral diversity and time across all individuals and gene regions simultaneously using a single, unified model. To do so, we model the relationship between time since infection and viral sequence diversity linearly, using all of the data from all individuals and gene regions, with a baseline slope term. Within this same model, we also include individual-specific and gene-region-specific slope-modifying terms to account for differences in slope for each individual and gene region relative to the baseline slope (Fig 1B). We will refer to this model as the *infant-trained hierarchical model*.

We designed this model such that it could use APD measures to predict time since infection for infants. As such, the inferred rates of viral diversification, or APD slopes, from this model will have units months/diversity which correspond to our predictive set-up. A similar approach was taken in a previous study estimating rates of viral diversification in an adult cohort [1]. For this reason, analysis and plots after Fig 1A will have pairwise diversity on the x axis, and time on the y axis.

The Bayesian modeling framework allowed us to incorporate existing biological knowledge when formulating the model. Because it has been shown that the majority of infants are infected with only a single viral variant [23–27], we restricted the baseline slope term to be a positive value in the model formulation using a prior, or an initial set of plausible values. However, because some individuals and gene-regions may accumulate viral diversity differently, we did not restrict the sign of the individual-specific and gene-region-specific slope-modifying terms. As such, it is still possible for the APD slope (i.e. the sum of the baseline slope, individual-specific slope-modifying term, and the gene-region-specific slope-modifying term) to be negative in value, which would suggest that diversity decreases with time for a given individual and gene region. Further, because we assumed that sequence diversity should be zero at infection time, we did not infer an intercept parameter.

## Viral diversity increases with time for all individuals and gene regions

Each viral sequence diversity (APD) measure represents the probability that two randomly drawn sequences have different nucleotides at a specified position, averaged over all positions, and the APD slope (i.e. rate of APD accumulation measured in units of months per diversity) represents the rate at which the APD measure increases. Because mutations will saturate within a sequence over time, APD slope should not be interpreted as the time required for all sites between two sequences to be mismatched. Instead, we suggest considering the inverse of APD slope, measured in units of diversity per month, which can be interpreted as the rate at which mismatches between two sequences accumulate. We estimated full posterior distributions of APD slopes from a sum of the baseline slope, individual-specific slope-modifying terms, and gene-region-specific slope-modifying terms using Hamiltonian Markov Chain Monte Carlo sampling from the *infant-trained hierarchical model*, given the full infant training data set (see Methods). These APD slope distributions are characterized by different medians and credible intervals (the Bayesian analogue of a confidence interval) for each individual and

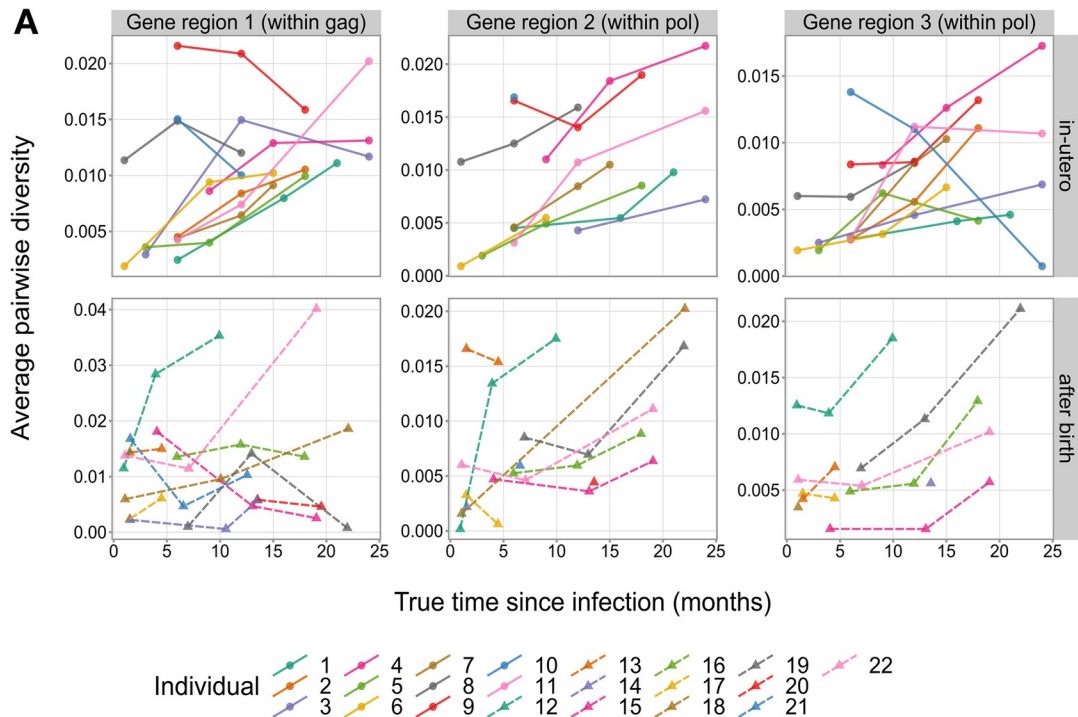

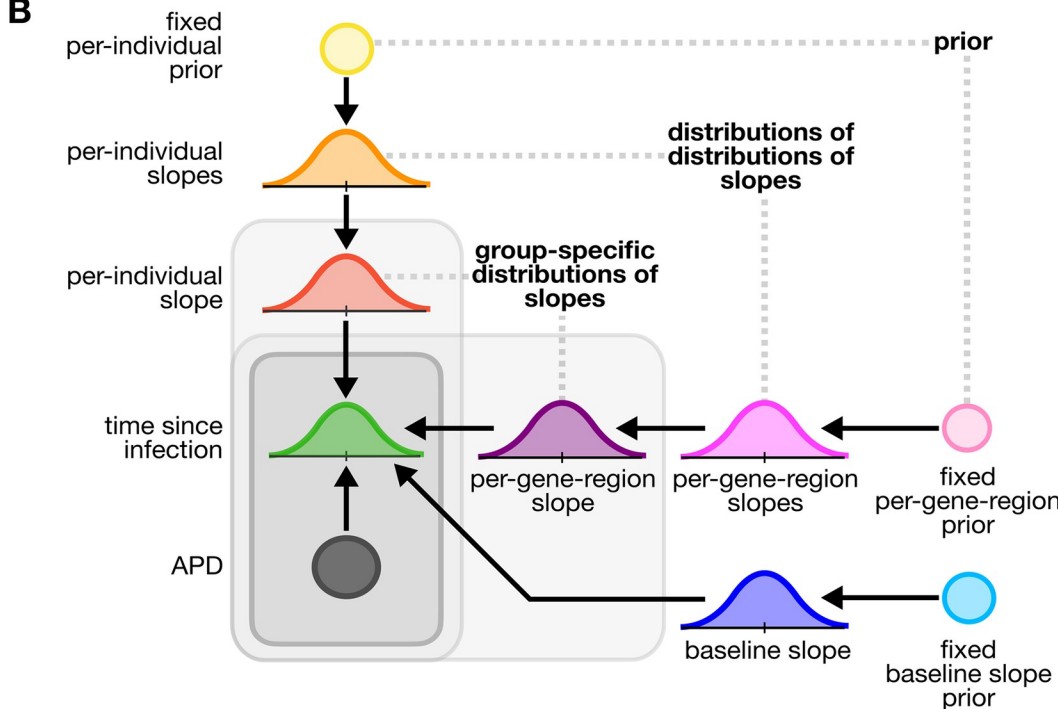

**Fig 1. Overview of data set and model used for quantifying rates of viral diversification. (A)** Average pairwise diversity (APD) as a function of true time since infection for each gene region within the data set. A separate line is shown for each individual in the data set. (**B**) We quantify rates of viral diversification for each individual using the *infant-trained hierarchical model*. The Bayesian hierarchical modeling framework used for this model allows us to obtain a distribution of estimated times since infection given a viral sequence diversity measure (APD) using a distribution of APD slopes which is obtained from a

distribution of baseline slopes in addition to individual-specific and gene-region-specific slope-modifying terms. Together, these distributions of APD slopes provide an estimate of the rates of viral sequence diversification for each individual and gene region. These APD slopes will have units months/diversity which correspond to our predictive set-up.

gene region. All median APD slopes were positive and only one credible interval contained negative values, suggesting that sequence diversity increases with time for all individuals and gene regions (S1 Fig). Overall, the inferred baseline slope suggested a positive rate of APD accumulation (median APD slope = 1262.16 months/diversity, 89% credible interval = [900.60, 1606.44]).

To assess the adequacy of the *infant-trained hierarchical model* for measuring APD slope, we did a series of posterior predictive checks which involve comparing the observed time since infection data to simulated samples generated from the posterior predictive distribution. This posterior predictive distribution is the model-derived distribution of possible times since infection for new observations given the observed data. Ideally, if the model fits the data well, then replicated time since infection data generated under the model should look similar to the observed time since infection data. Indeed, this is mostly what we observe (S3 Fig). However, these replicated data contained a wider range of time since infection values suggesting that the model allows for more variation in APD slope and possible time since infection than expected, given the observed data.

## Rates of viral sequence diversification vary by individual, gene region, and mode of infection, but not by set-point viral load or rate of CD4+ T cell decline

We were interested in whether rates of viral sequence diversification (i.e. APD slope) varied significantly by individual, gene region, mode of infection, set-point viral load, or rate of CD4 + T cell decline. To explore whether APD slopes varied by individual or gene region, we compared the *infant-trained hierarchical model* (which contains a baseline slope term, as well as individual-specific and gene-region-specific slope-modifying terms) to models lacking the respective slope-varying effects of interest (e.g. the individual-specific slope-modifying term or the gene-region-specific slope-modifying term) by calculating Bayes factors. Similarly, to explore whether APD slopes varied by mode of infection, set-point viral load, or rate of CD4 + T cell decline, we compared the *infant-trained hierarchical model* to models containing the respective slope-varying effects of interest (e.g. a mode-of-infection-specific slope-modifying term, a set-point-viral-load-specific slope-modifying term, etc.) by calculating Bayes factors. Within the Bayesian framework, Bayes factors provide a method for comparing two models and measuring the weight of evidence in favor of one model relative to the other. The Bayes factor test assigns different interpretations depending on the magnitude of the resulting Bayes factor value (S1 Table) [35]. Here, a Bayes factor greater than 100 provides extremely strong evidence in favor of the *infant-trained hierarchical model* and a Bayes factor less than 1/100 provides extremely strong evidence in favor of the alternative model which lacks or contains the slope-modifying effect of interest. As such, both large and small Bayes factors (BFs) contain useful information.

With these methods, we separately compared the hypotheses that APD slopes vary by gene region or individual against the alternative hypotheses that the rates do not vary. In both cases, we found extremely strong evidence in favor of the *infant-trained hierarchical model*, suggesting that APD slopes varied both across individuals (BF = $6.710 \times 10^{20}$) and gene regions (BF = $3.963 \times 10^4$). These trends were reflected when looking closer at the data (Fig 1A) and

the inferred values of the individual-specific and gene-region-specific slope-modifying terms from the *infant-trained hierarchical model* (Figs 2 and S1). The inferred individual-specific slope-modifying terms, which model the differences in APD slope for each individual relative to the baseline slope, suggested that APD slopes vary substantially across individuals (range of median slope difference [89% credible interval] = -873.48 [-1247.16, -518.64] to 934.08 [445.32, 1451.16] months/diversity). The inferred gene-region-specific slope-modifying terms, which model the differences in APD slope for each gene region relative to the baseline slope, suggested that the APD slope accumulated faster within gene region 1 (within *gag*; median slope difference = -181.32 months/diversity, 89% credible interval = [-464.40, -85.68]) compared to gene region 2 (5' end of *pol*; median slope difference = 7.20 months/diversity, 89% credible interval = [-271.92, 299.64]) and gene region 3 (3' end of *pol*; median slope difference = 191.04 months/diversity, 89% credible interval = [-49.08, 525.72]).

We were also interested in whether APD slopes varied by timing of infection, specifically comparing infants who tested positive for HIV at birth and were infected in-utero to infants who first tested positive for HIV after birth. We will refer to this relative timing of infection as "mode of infection." As such, we compared the *infant-trained hierarchical model* to an

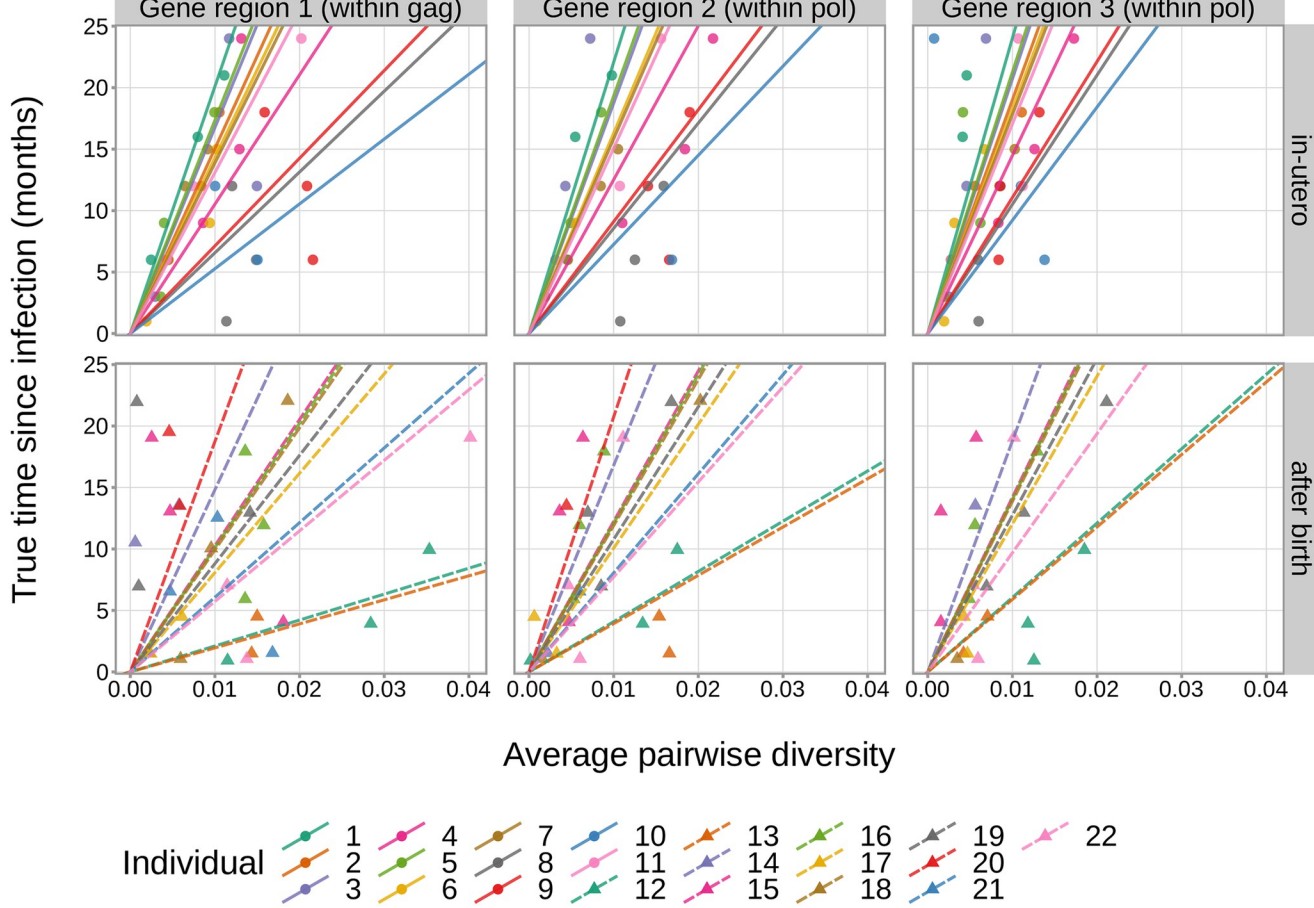

**Fig 2. APD slopes vary by individual, gene region, and mode of infection.** A different line is shown for each individual and gene region in the training data set. The slope of each line is the *infant-trained hierarchical model*-derived median APD slope for each individual and gene region. Since the model we are using to estimate these APD slopes was designed to eventually be used for estimating time since infection given an APD value, these inferred APD slopes have units months/diversity which correspond to our predictive set-up.

alternative model containing a mode-of-infection-specific slope-modifying term. For this comparison, we found extremely strong evidence in favor of the alternative model suggesting that APD slope varied by mode of infection (BF = $1.199 \times 10^{-4}$). Specifically, the mode-of-infection-specific slope-modifying term, which models the difference in APD slope between modes of infection relative to the baseline slope, suggested that individuals infected after birth had a higher rate of APD accumulation (median slope difference = -88.44 months/diversity, 89% credible interval = [-541.20, 253.08]) compared to individuals infected in-utero (median slope difference = 116.76 months/diversity, 89% credible interval = [-146.4, 674.88]). These differences can be seen visually in Fig 2. A reviewer suggested that this signal was driven by the two postpartum-infected individuals (individuals 12 and 13) who had relatively higher rates of viral diversification compared to the other postpartum-infected individuals. Indeed, we found that APD slope did not vary substantially with mode of infection (BF = 2.203) when these two individuals were excluded from the analysis. Regardless of whether the variation in APD slopes by mode of infection represents a true biological signal, we chose not to use the alternative model that includes a mode-of-infection-specific slope-modifying term for downstream analyses. This decision was made because we do not expect to have knowledge of the mode of infection when applying our final model to individuals with unknown infection timing.

Lastly, because set-point viral load and rate of CD4+ T cell decline are both strong predictors of disease progression and survival [3,31,36–42], we were interested in whether APD slopes varied in the context of these variables. We expected that individuals with higher set-point viral loads (i.e. higher rates of viral replication, which is error-prone) would have higher rates of viral diversification. Likewise, we expected that individuals with higher rates of CD4 + T cell decline (i.e. faster disease progression) would also have higher rates of viral diversification. To explore this, we compared APD slope to set-point viral load (Fig 3) and to rate of CD4 + T cell decline (S2 Fig). Despite our initial hypotheses, there did not appear to be a clear trend in either context. To quantify this, we compared the *infant-trained hierarchical model* to alternative models containing either an interaction slope term between APD and log10 of set-point viral load or an interaction slope term between APD and rate of CD4+ T cell decline using a Bayes factor test. In both cases, we found moderate evidence in favor of the *infant-trained hierarchical model*. Our conclusion is that APD slope does not vary substantially with log10 of set-point viral load (BF = 3.484) or rate of CD4+ T cell decline (BF = 4.840).

## Infant time since infection is estimated most accurately using an infant-trained model

Given that APD appears to increase with time during pediatric infection, we were interested in determining whether viral APD could be used to accurately estimate time since infection in infants. To explore this, we evaluated three different model types.

A previously-published adult study has shown that rates of viral diversification can be used to accurately estimate infection timing using least absolute deviation linear regression models [1]. Specifically, they define a unique model for each viral gene region in which diversity is measured such that each model takes a viral diversity measure from that gene region as input and outputs an estimated time since infection. They train each model using an adult data set. We refer to this group of existing models as the *adult-trained linear models*. Here, we have replicated these methods to train a least absolute deviation linear regression model for each viral gene region using the infant training data set described above (see Methods). We will refer to these models as the *infant-trained linear models*. Both the *adult-trained linear models* and the *infant-trained linear models* are summarized in Table 1. Given that there are three gene region regions represented in our infant training data set, the groups of *adult-trained linear models*

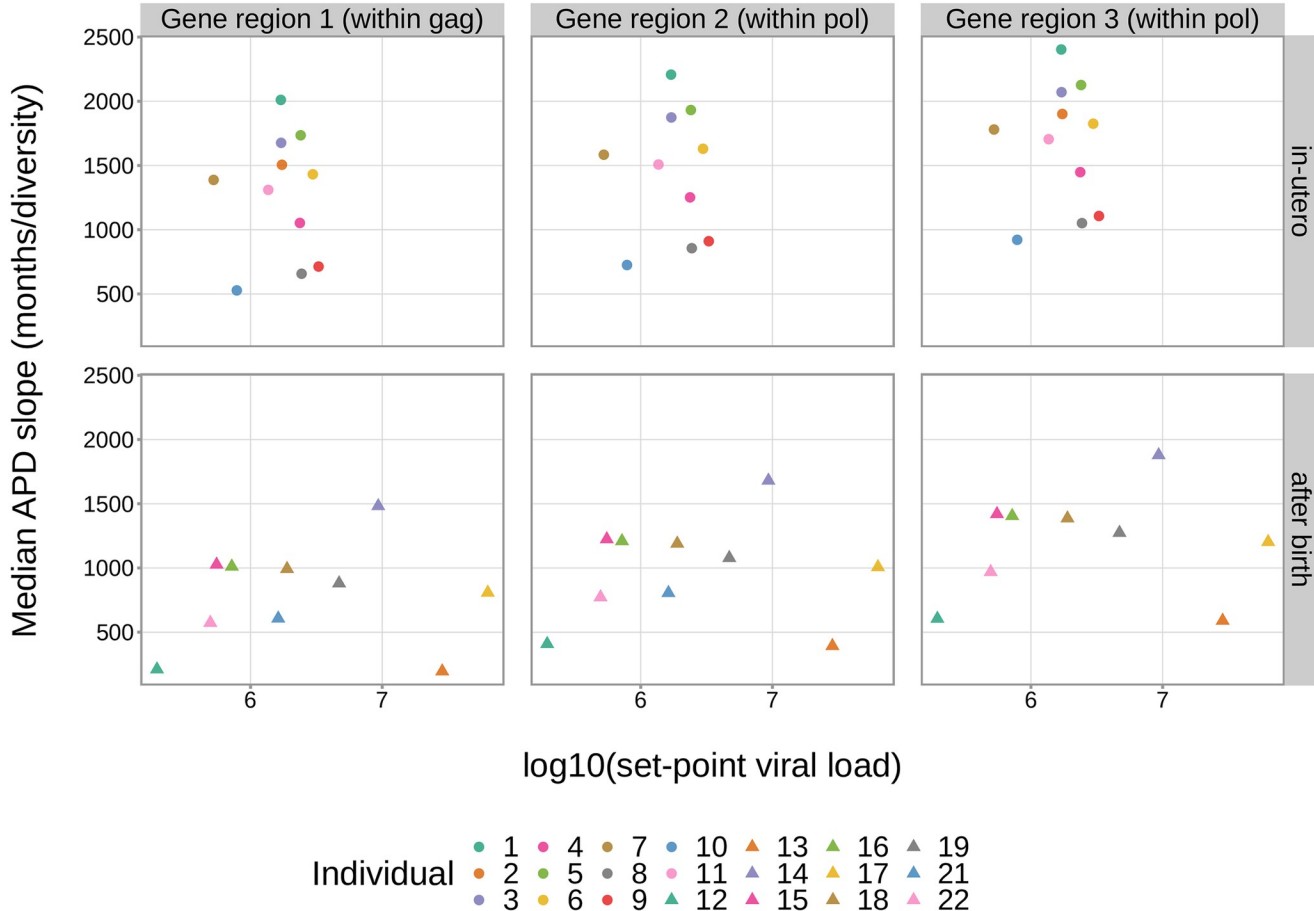

**Fig 3. APD slope does not vary substantially by log10 of set-point viral load.** The inferred median APD slope for each individual and gene region from the *infant-trained hierarchical model* is shown as a function of the log10 of set-point viral load for each individual.

**Table 1. Description of models.**

| Model name | *Infant-trained hierarchical model* | *adult-trained linear models* | *infant-trained linear models* |
|---|---|---|---|
| **Model type** | Bayesian hierarchical regression model | Least absolute deviation linear regression model | Least absolute deviation linear regression model |
| **Input** | Single viral diversity measure (sampled from any of the three gene regions) | Single viral diversity measure (sampled from the gene region specific to the model being used) | Single viral diversity measure (sampled from the gene region specific to the model being used) |
| **Output** | Distribution of estimated times since infection | Point estimate of time since infection | Point estimate of time since infection |
| **Data set used for model training** | Infant training data set presented here | Adult training data set [1] | Infant training data set presented here |
| **Includes individual- and gene-region-specific slope-modifying terms** | Yes | No | No |
| **Total model count** | 1 | 3 (one for each gene-region) | 3 (one for each gene-region) |

and the *infant-trained linear models* each consist of three separate models (e.g. one for each unique gene region).

All of these simple linear regression models assume that viral diversity accumulates identically across individuals, and provide only a single estimated infection time for a given viral sequence diversity measure. In contrast, the Bayesian *infant-trained hierarchical model* (described earlier) goes further by affording us the flexibility of obtaining a single, unified model using sequence diversity measures sampled from multiple time points for each individual and gene region in the infant training data set (Fig 1B). Further, the Bayesian modeling framework gives us the ability to assess the probability of all possible infection times for any given viral sequence diversity measure by estimating a full posterior distribution over model parameters, given the infant training data set (see Methods).

We were interested in how these three model types (Table 1) compared in their ability to accurately predict time since infection for the infant data used here. To measure the accuracy of each model, we compared the model-derived time since infection to the "true" time since infection obtained using frequent HIV testing for each individual. Because the *infant-trained hierarchical model* and the *infant-trained linear models* were all fit using the infant training data set described above, we used leave-one-out-cross-validation to obtain the model-derived time since infection estimate for each individual and APD measure within the infant training data set ("infant-trained model evaluation paths" in Fig 4). In contrast, because the coefficients from the *adult-trained linear models* were fit using a separate adult dataset within a previous study [1], we used these models to obtain model-derived time since infection estimates for each individual and APD measure within the infant training data set directly ("adult-trained model evaluation paths" in Fig 4).

While the *infant-trained hierarchical model* provides a distribution of estimated times since infection for each viral sequence diversity measure, the *infant-trained linear models* and the *adult-trained linear models* only infer a single point estimate of the time since infection for each viral sequence diversity measure. As such, to directly compare models, we treated the median of each distribution of estimated times since infection from the *infant-trained hierarchical model* as an equivalent measure to the time since infection point estimate from the other models. To calculate the accuracy of each model, we compared each model-derived time since infection estimate to the HIV-testing-derived "true" time since infection measures for each

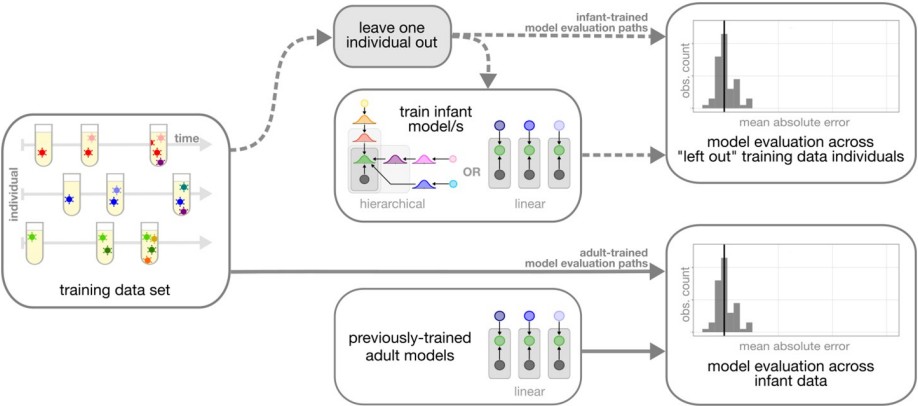

**Fig 4. Overview of model evaluation strategy.** We trained each infant-trained model using subsets of the infant training data set. We used leave-one-out-cross-validation to evaluate the prediction accuracy of each infant-trained model on the training data set (shown by the dashed lines). We also tested the prediction accuracy of the previously-published *adult-trained linear models* using the infant training data set (shown by the solid lines).

individual and computed a per-gene-region mean absolute error (MAE). With these methods, we found that all three model types had their lowest prediction errors using APD measures sampled from gene region 3 (3' end of *pol*), followed by gene region 2 (5' end of *pol*) (Fig 5A). While the *infant-trained linear models* had the lowest prediction errors across all gene regions, the MAEs were very similar for the *infant-trained hierarchical model* and the *infant-trained linear models* (e.g. within gene region 3, MAE = 5.698 months and 5.031 months, respectively) and their distributions of prediction errors were tightly peaked around zero (Fig 5A). Both infant-trained models had much lower prediction error compared to the *adult-trained linear models* (e.g. within gene region 3, MAE = 20.252 months). We also designed a version of the *infant-trained hierarchical model* in which observations were drawn from a Laplace distribution, corresponding to the least absolute deviation assumption (see Methods), and noted very similar results (S4 Fig).

Given that the mean absolute error estimates for both infant-trained models were highly similar, we wanted to more closely compare the prediction accuracy of the two models. When evaluating the relationship between true time since infection and model-derived time since infection values, we found that the two values were slightly more correlated for the *infant-trained hierarchical model* compared to the *infant-trained linear models* for gene region 1 (within *gag*; $R^2$ = 0.018 versus 0.003), gene region 2 (5' end of *pol*; $R^2$ = 0.185 versus 0.114), and gene region 3 (3' end of *pol*; $R^2$ = 0.124 versus 0.112) (Fig 5B). Likewise, when quantifying the relationship between true time since infection and model-derived time since infection values, we found that the slope for the line of best fit was closer to one for the *infant-trained hierarchical model* compared to the *infant-trained linear models* for gene region 1 (within *gag*; $\beta$ = 0.172 versus -0.014), gene region 2 (5' end of *pol*; $\beta$ = 0.469 versus 0.213), and gene region 3 (3' end of *pol*; $\beta$ = 0.352 versus 0.203). For both models, prediction error depended on true time since infection; on average, model-derived time since infection was overestimated for times since infection that were less than 11.184 and 9.936 months for the *infant-trained hierarchical model* and *infant-trained linear models*, respectively (Fig 5C). Likewise, model-derived time since infection was underestimated for times since infection that were greater than 11.184 and 9.936 months for the *infant-trained hierarchical model* and *infant-trained linear models*, respectively. This trend was slightly less strong for the *infant-trained hierarchical model*.

To further evaluate the prediction accuracy of the *infant-trained hierarchical model*, we calculated the posterior predictive coverage of the model, which provides a measure of the proportion of observed time since infection values that fall within their model-derived time since infection posterior prediction intervals. Said more simply, it is how frequently the true values fall within the "error bars" displayed in the top row of Fig 5B. For this calculation, we obtained the model-derived time since infection posterior prediction interval for each individual and APD measure using leave-one-out-cross-validation. With these methods, we found that 90.96% of all true time since infection measurements fell within the 89% credible interval of their model-derived time since infection posterior distribution. This high coverage probability suggests that the *infant-trained hierarchical model* has well-calibrated uncertainty predictions which allows us to understand the confidence of a given prediction. However, the width of the 89% credible interval for each model-derived time since infection posterior distribution is 17.52 months. This lack of precision underlines the uncertainty that is inherent to predictions using these data, despite allowing for variation between individuals, sequencing regions, etc., and further explains the limited prediction accuracy observed for all predictive methods analyzed here.

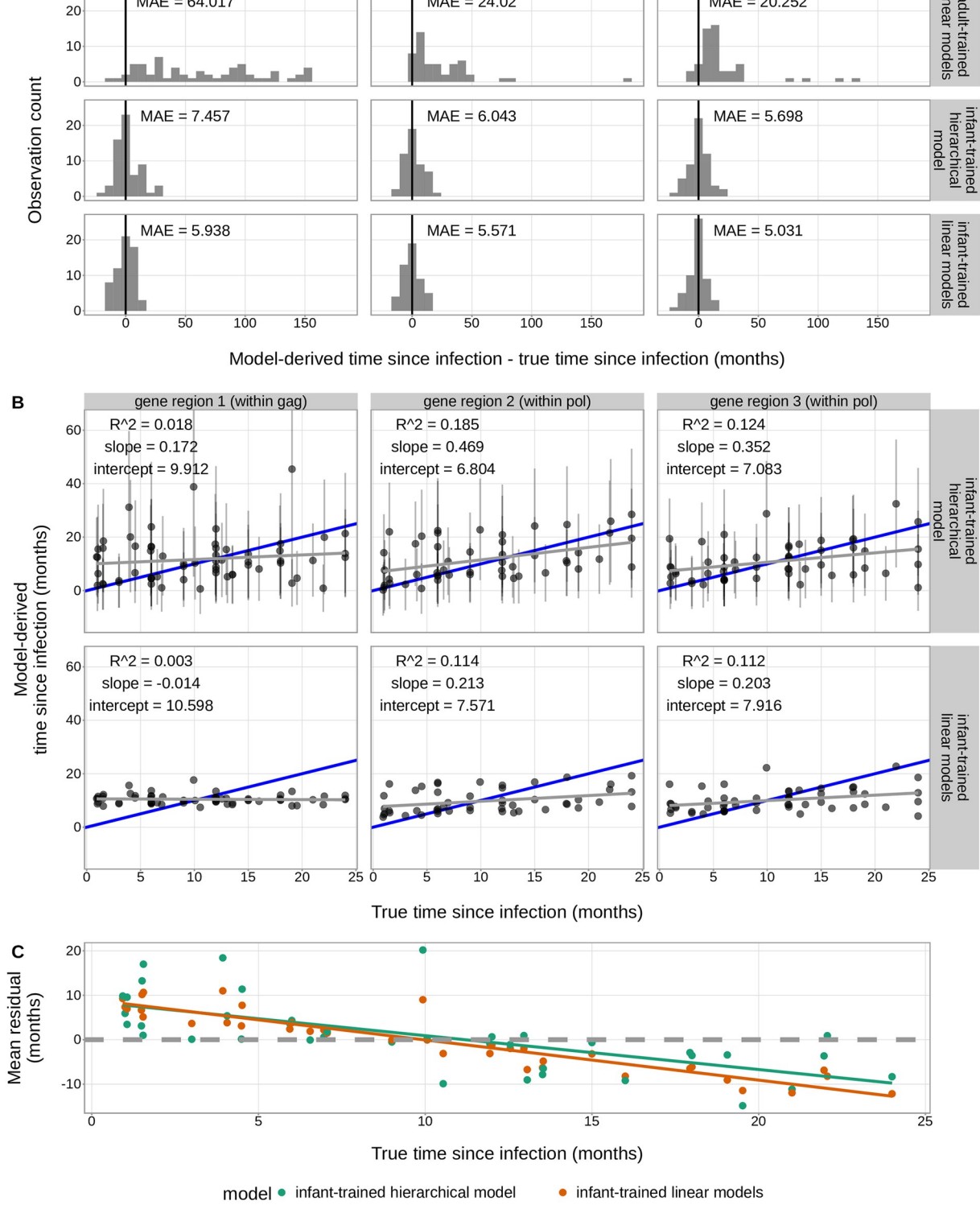

**Fig 5. Comparison of model prediction errors for the training data set.** As previously described, the *infant-trained hierarchical model* is trained using viral diversity measures from all gene regions (but takes only a single viral diversity measure as input, Table 1) whereas for the *infant-trained linear models* and the *adult-trained linear models* there is a unique model trained for each gene region. For all models, prediction errors are computed separately for each gene region. For both infant-trained model types, leave-one-out-cross-validation was used to calculate the estimated time since infection for each individual. (**A**) Both infant-trained models had much lower mean absolute error compared to the

*adult-trained linear models*. The difference between model-derived time since infection and true time since infection is shown on the x-axis. A difference of zero is shown by the black vertical line. (**B**) Comparison of true time since infection and model-derived time since infection for the two infant-trained models. For the *infant-trained hierarchical model*, the 89% credible interval is shown as a vertical black line for each estimate. The blue line represents the model-derived time since infection = true time since infection line. The gray line shows the line of best fit between model-derived time since infection and true time since infection. (**C**) For both infant-trained models, the mean residual (given by the average difference between true time since infection and model-derived time since infection) changes as a function of true time since infection. The mean residual for each recorded time since infection is plotted for each model: green = *infant-trained hierarchical model*, orange = *infant-trained linear models*. A mean residual of zero is shown by the horizontal dashed line.

## Validation using an independent data set

We validated the inferred coefficients from each model type on an independent testing data set from 15 infants with well-defined infection timing and sequencing data from a single time point. Before analyzing these data, we "froze" our trained model coefficients (for the *infant-trained linear models*) and posterior distributions over model parameters (for the *infant-trained hierarchical model*) in git commit 1f6652a on our repository (see https://github.com/matsengrp/infection-timing). Because the regression coefficients from the *adult trained linear models* were previously-published, they were also "frozen" prior to the analysis. For each model, we then inferred the time since infection for each of the 15 infants in the testing data set using the frozen regression coefficients and computed a per-gene-region mean absolute error (MAE) (Fig 6).

These new testing data exhibited slightly different behavior from the training data set for all three models. As before, using viral diversity measures sampled from gene regions 2 and 3 (both within *pol*), we found that the MAEs were highly similar for the *infant-trained hierarchical model* and the *infant-trained linear models* (within gene region 2 (5' end of *pol*), MAE = 9.880 months and 10.241 months, respectively, and within gene region 3 (3' end of *pol*), MAE = 9.452 months and 9.067 months, respectively) (Fig 7). However, the *adult-trained linear models* had relatively lower error using diversity measures sampled from gene region 3 (3' end of *pol*) (MAE = 8.640 months). While prediction errors were highest for all three models when using viral diversity measures sampled from gene region 1 (within *gag*) from the training data set, for these testing data, the prediction error was lowest using viral diversity

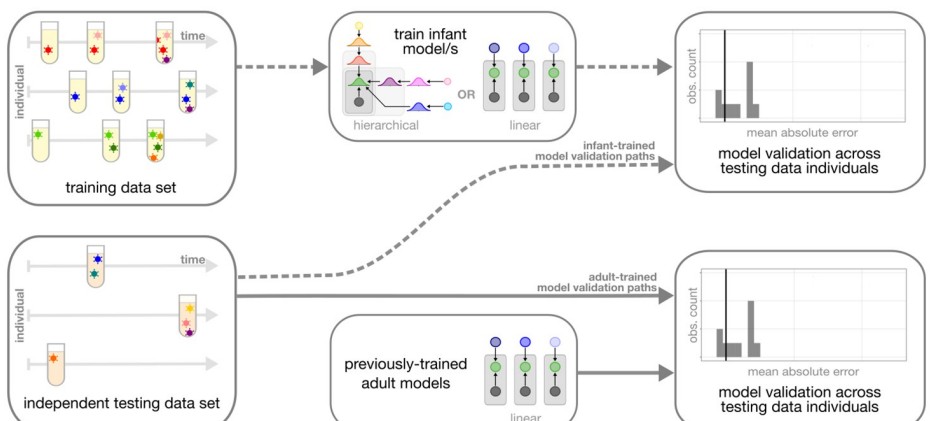

**Fig 6. Overview of model validation strategy.** We trained each infant-trained model using the full infant training data set and "froze" the coefficients. The model coefficients for the *adult-trained linear models* were previously-published. For each model, we validated the prediction accuracy on an independent testing data set.

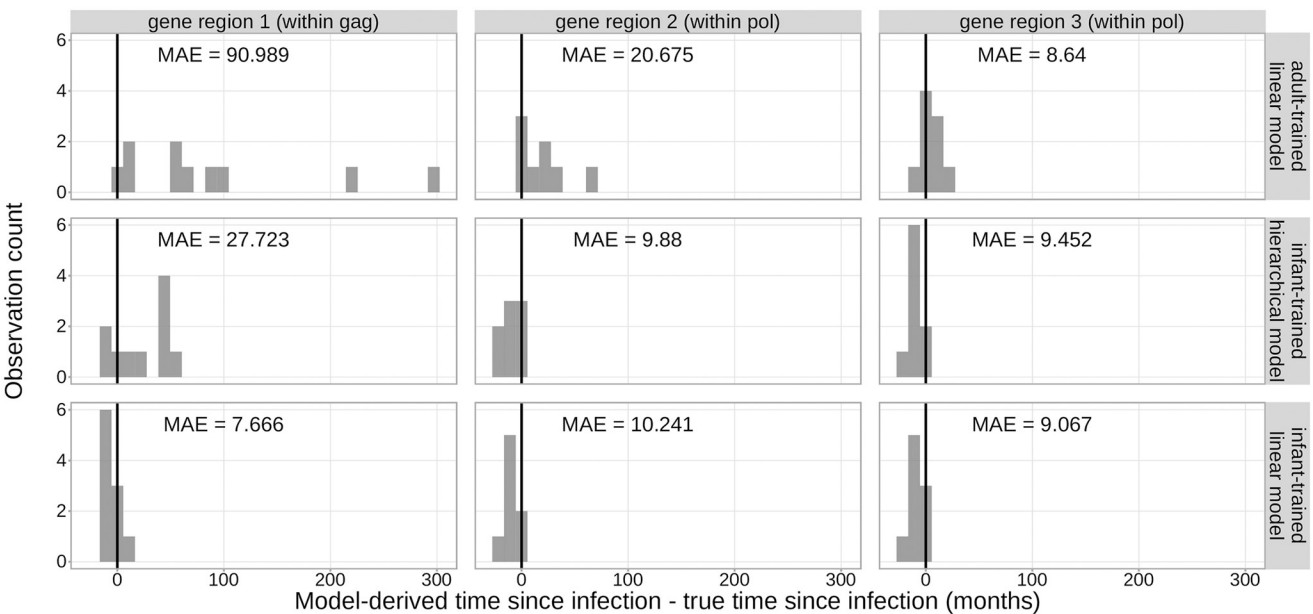

**Fig 7. Comparison of model prediction errors by gene region for the testing data.** The difference between model-derived time since infection and true time since infection is shown on the x-axis. A difference of zero is shown by the black vertical line.

measures sampled from gene region 1 with the *infant-trained linear models* (MAE = 7.666 months). Using these same gene region 1 diversity measures yielded high error with the *infant-trained hierarchical model* and the *adult-trained linear models* (MAE = 27.723 months and 90.989 months, respectively). We believe these error differences could be attributed to a difference in properties between the two data sets.

Indeed, we noted that the distributions of APD measures and times since infection for these 15 testing cohort infants were different from the training cohort of 22 infants. For example, the APD measures from gene region 3 (3' end of *pol*) for this testing data set were slightly lower and the APD measures from gene region 1 (within *gag*) were much higher compared to the training data set (S5 Fig). Further, the true times since infection for this testing data set were also much higher compared to the training data set (S6 Fig). These differences in properties likely explain some of the differences in prediction errors between the training and testing data sets for each model (compare Figs 5A to 7).

## Discussion

Knowledge of viral diversity dynamics and timing of HIV acquisition are essential for improving our understanding of viral pathogenesis and epidemiology of pediatric HIV infection. While adult studies have characterized rates of viral diversification and established that they can be used to accurately estimate adult infection timing, rates of viral diversification have yet to be explored in infants. Using Illumina sequencing of *gag* and *pol* from longitudinal plasma samples from a cohort of Kenyan infants with well-defined infection timing, we have (1) quantified viral diversity dynamics during the early stages of pediatric HIV infection and (2) demonstrated that infant time since infection can be estimated using <u>infant-specific</u> rates of viral diversification, but not adult-specific rates.

Given that infants are often slower to suppress virus [31], infected with viruses possessing relatively higher replicative fitness [29,30], and have higher HIV set-point viral load levels [2,3]

compared to adults, it has been previously suggested that rates of viral diversification likely differ between the two groups [32]. Indeed, a previous adult study reported that viral diversity increases at a rate of 0.00024 diversity/month (adult model slope of 4206.0 months/diversity) using sequences sampled from the *pol* gene within an adult cohort [1]. Here, using equivalent *pol* gene viral diversity measures from an infant cohort, we showed that viral diversity increases at a median rate of 0.00079 diversity/month (infant model slope of 1269.6 months/diversity). This finding suggests that viral diversity accumulates much faster during pediatric infection relative to adult infection and highlights the importance of considering these differences when developing methodologies for future studies related to HIV infection timing across different age groups. Depending on the study, failure to appropriately account for accelerated rates of HIV sequence evolution in infants could result in erroneous conclusions regarding the timing and/or source of pediatric infections.

Further, we found that these infant viral diversity dynamics varied by gene region such that viral diversity accumulated faster within *gag* compared to *pol*. These observations are consistent with previous adult studies [1]. We also reported that rates of viral diversification varied substantially across individual infants which may be related to both host and viral factors. It is possible that individuals infected with more than one viral variant could experience a decrease in viral diversity during the early stages of infection as a result of selection effects, however, infant infections are most often initiated by a single viral variant [23–27]. In fact, we found that only one individual (individual 10) had a decrease in viral diversity over time across all gene regions.

Interestingly, we found that rates of viral diversification varied significantly by timing of infection such that individuals who tested positive for HIV after birth had a higher rate of viral diversification compared to individuals who tested positive for HIV at birth and were infected in-utero. This signal appeared to be driven by two postpartum-infected individuals (individuals 12 and 13) who had relatively higher rates of viral diversification compared to the other postpartum-infected individuals. In fact, when we removed these individuals from our analysis, we no longer found a difference in viral diversification rates between the two groups. We do not rule out that these are instead cases of very late in utero infections, which may not test positive at birth. Regardless, this result is surprising given that in-utero-infected individuals have been previously shown to have relatively higher HIV set-point viral load levels and faster disease progression compared to individuals infected after birth [2–4,40]. We expected that individuals with higher set-point viral load levels and higher rates of CD4+ T cell decline, and thus higher rates of viral replication and faster disease progression, would have higher rates of viral diversification. However, we found that rates of viral diversification did not vary substantially with set-point viral load levels or rate of CD4+ T cell decline. It is possible that the limited number of samples included in this analysis may have restricted our ability to detect a signal. Because this analysis included a limited range of set-point viral load levels and rates of CD4+ T cell decline across the individuals in this study, further work will be required to explore the relationship between viral diversification rates and set-point viral load levels/rate of CD4+ T cell decline in infants.

Adult studies have shown that linear regression models using viral diversity can be used to estimate adult infection timing [1,18]; These adult-specific models report mean prediction errors ranging from 10.08–12 months [1,18] across independent adult cohorts. Here, we explored the potential for using this approach to estimate infection timing in infants. We found that infant time since infection can be estimated more accurately using infant-trained models compared to previously-developed adult-trained models [1]. Specifically, we found that infant-trained linear regression models, which were trained using similar methods to the previously-developed adult-trained models, led to the most accurate estimates across two

independent infant data sets. An infant-trained Bayesian hierarchical regression model which accounted for individual- and gene-region-specific variations in viral diversification rates had similar performance. The large differences in viral diversification rates between infants and adults described above likely explain why infant-trained models lead to more accurate estimates of infant infection timing compared to adult-trained models. Despite this increased accuracy, we still found a lack of precision across our best infant-trained models (prediction error from all gene region 3 (3' end of *pol*) APD measures with the *infant-trained linear model* range = 0–19.8 months, mean = 5.6 months) which underlines the uncertainty that is inherent to predicting infection timing using diversity measures.

For both infant-trained model types, viral diversity sampled from within the *pol* gene led to the most accurate infection timing estimates compared to *gag*. This suggests that viral diversity may accumulate more linearly within *pol* compared to *gag* during the early stages of infant infection. This finding is consistent with previous studies of viral diversity accumulation during adult infection [1]. However, for both infant-trained models, we reported that prediction error depended on the true time since infection such that true time since infection values less than ~12 months were overestimated by each model and true time since infection values greater than ~12 months were underestimated by each model. We noted even more extreme results when validating each infant-trained model using the infant testing dataset which contained data from infants whose true time since infection values were relatively larger compared to the training dataset. These trends suggest that viral diversity may not accumulate in a perfectly linear fashion during the early stages of pediatric infection. Instead, it appears that viral diversity may accumulate faster during the first year of infection relative to the second year of infection. This may be possible if, for example, rapidly increasing viral load levels during very early infection result in relatively higher rates of viral replication/diversification, perhaps as a result of viral adaptation to a new human host, compared to when set-point viral load levels are established later on. Given that infants are infected in the presence of maternal antibodies, another possible explanation is that pressure from maternal passive antibodies may lead to an increase in viral replication/diversification during the early stages of infection relative to later stages. Further work consisting of more frequent diversity sampling during early infection will be required to explore these relationships and formulate appropriate regression models (i.e. nonlinear models, models responsive to viral population size, etc.).

Here we developed and evaluated methods for estimating infection timing that are suitable for short, unlinked sequencing reads, a data type used in some adult-specific models [1]. Other types of sequencing data enable additional infection timing methodologies. For example, given single genome sequence data, one can use BEAST, a Bayesian phylogenetic software, to conduct a Bayesian phylogenetic analysis for molecular dating. This approach has been previously shown to accurately estimate infection timing of adult HIV infections [43].

There are several key limitations of our study. First, because we were interested in quantifying viral diversity dynamics during pediatric HIV infection prior to antiretroviral therapy, our study relied on the availability of banked plasma samples from infants who were diagnosed with HIV before antiretroviral therapy was standard of care (e.g. infants diagnosed in Kenya between 1992–2002) whose infection timing was well-defined. Next, while all of the infants included in this study were naive to antiretroviral therapy for the duration of their monitoring, many of their mothers received a short course of zidovudine (AZT) during pregnancy, which was standard of care in Kenya at the time of cohort enrollment to reduce the risk of mother-to-child transmission. In fact, for the infants included in the training cohort, only 3 out of 22 mothers did not receive short-course AZT during their pregnancies. While it is possible that this lack of treatment could have influenced the rates of viral diversification for the infants born to these mothers, we did not have the statistical power to explore this relationship due to

the small number without AZT. Also, we noted several differences in properties between the training and testing data sets which may explain the differences in prediction errors between the two data sets when using each model. For example, the three testing data set observations with the largest over-prediction errors from all of the infant-trained models each had gene region 1 (within *gag*) APD measures that were greater than 0.04 and times since infection that were greater than 15.6 months. Observations with similar gene region 1 APD measures and times since infection appeared at very low frequencies within the training data set. Next, because our infection timing models were formulated using the assumption that the majority of infants are infected with only a single viral variant [23–27], they may produce inaccurate time since infection estimates for individuals with multi-founder infections. Lastly, since the infection timing for each individual was defined using HIV testing intervals between 6–12 weeks, the known infection time of each individual contains errors that limited the accuracy of our infection time modeling.

In summary, we have found that viral diversity increases over time during the first two years of pediatric infection and that infant-specific rates of viral diversification can be used to estimate pediatric infection timing. These results further our understanding of pediatric HIV pathogenesis and how it may differ from adult infection, particularly in regards to differences in rates of HIV sequence diversification.

## Materials and methods

### Ethics statement

The CTL and NBT sub-studies described here were approved by the Kenyatta National Hospital–University of Nairobi Ethics and Research Committee, the Fred Hutch Cancer Center Institutional Review Board, and the University of Washington Institutional Review Board. Written informed consent was obtained prior to enrollment from the child's parent/guardian, which included use of the child's samples in future studies.

### Study participant selection

Our study utilized banked plasma specimens from CWH enrolled in two Kenya-based cohorts to train and validate each model. The infant training data set were derived from banked longitudinal plasma samples from 22 infants enrolled in the Cytotoxic T Lymphocyte (CTL) cohort in Nairobi between 1999–2002 [34] and the infant testing data set included 10 additional infants from the CTL cohort and 5 infants from the Nairobi Breastfeeding Trial (NBT) conducted between 1992–1998 [33]. These studies enrolled women living with HIV (LWHIV) late in pregnancy and followed the mother-infant pairs for 1–2 years following delivery.

Both CTL and NBT studies were conducted prior to the widespread availability of antiretroviral therapy (ART), so all infants in this study were ART-naive for the duration of monitoring.

### Estimating true HIV infection time

Infants were tested for HIV DNA at birth, 6 weeks, 14 weeks, and every 3 months until 2 years of age. To more precisely estimate time of infection, samples prior to the first HIV-DNA-positive test were tested for HIV RNA. Time of infection was estimated as the midpoint between the last negative HIV DNA or RNA test and the first positive test. Twenty-two infants had multiple timepoints available and were included in the training data set, of which 11 were HIV-positive at birth and 11 were HIV-negative at birth and subsequently tested HIV-infected after 3 months of followup. Of the 15 additional infants (n = 10 from CTL and n = 5 from

NBT), 2 were HIV-positive at birth and 13 were HIV-negative at birth but tested positive subsequently. The estimated infection time measures were used in the models as described below. For individuals infected in-utero, time of infection was estimated as birth since HIV testing could not be conducted prior to delivery (i.e. in-utero). As such, these infection time measures contain uncertainty. To reduce this uncertainty for in-utero-infected individuals during some of our model training, we estimated the true infection time quantity directly in the modeling framework of the *infant-trained hierarchical model* described below.

## HIV-1 amplification and sequencing

Total RNA was extracted from 50uL of plasma using a previously described quanidine-based RNA extraction protocol [44]. HIV RNA from each sample was reverse transcribed (Superscript III First Strand Synthesis 18080051) following the manufacturer's protocol in three separate reactions using reverse primers located within the *gag* gene region (gene region 1) (HXB2 2347–2324), the middle of the *pol* region (gene region 2) (HXB2 3892–3869), and the 3' end of *pol* (gene region 3) (HXB2 5105–5077). Synthesized cDNA from the 3' end of *pol* (gene region 3) was quantified via an in-house ddPCR protocol (ddPCR TM Supermix for Probes, Bio-Rad 1863026) which used custom cross-subtype *pol* primers and probe as previously described [45]. Using HIV cDNA copies quantified by the ddPCR, we estimated the input of RNA copies of each sample per reverse-transcription reaction (RT) and normalized the cDNA template input into 2 replicate amplification PCRs to approximately equate 8,000 copies HIV RNA per RT-PCR reaction. Three PCR reactions using KOD Hot Start Master Mix (Millipore-Sigma 71842) were completed to amplify three gene regions: gene region 1 (HXB2 801–2252), gene region 2 (HXB2 1998–3799), and gene region 3 (HXB2 3681–5061). Amplification primers were adapted from previously described primers [46]. Gene regions 1 and 2 were single round amplifications using 35 cycles while gene region 3 required nested PCRs of 20 and 15 cycles (see S2 Table).

Prior to building sequencing libraries, all PCR amplicons were quantified using TapeStation D5000 Assay (100–5000 bp) (Agilent 4200) then normalized such that 0.5 ng of DNA was used for tagmentation and Nextera-XT DNA library preparation kit (Illumina, FC-131-1096) following manufacturer's protocols. Libraries were sequenced on the Illumina MiSeq v3 platform with 2x300bp sequencing kits (MS-10-3003) with a median number of 339,297 (IQR: 167,531–497,700) reads per amplicon.

## Sequence alignment and diversity measures

Raw sequencing reads are aligned to the reference HXB2 (*gag* or *pol*, depending on the fragment) using probabilistic alignment pipeline, HIVMMER [47]. Consensus sequences for each sample were assembled into a local database used to re-align all raw reads using Magic-BLAST [48] to filter for "contaminating" reads, or sample reads for a participant that aligned most closely to the consensus sequence of a different participant's sample. Filtered reads were aligned using HIVMMER a second time to generate new consensus sequences and resulting amino acid variant calls were used to calculate average pairwise diversity (APD) for each sample if > = 5000 read coverage was achieved. APD is a measure of the probability that two randomly selected sequences contain differing nucleotides at the third codon position averaged across the length of the sequence [1]. For this analysis, APD was calculated using only sites at which the sum of all minor variants was greater than 0.01.

## Training data set description

The model training data set consisted of data from 22 infants as described above. For each infant, we calculated viral APD measures for 2–3 timepoints within 3 different gene regions. We excluded data when, for a given individual and gene region, we only had one sampled time point. With the APD measures for each individual, gene region, and time point in the data set, we calculated the average APD value across sample sequencing replicates to get the final APD value used in model fitting and other analyses.

## Infant-trained hierarchical model overview

We want to infer time since infection (in years) using measures of viral sequence diversity. We use a Bayesian hierarchical modeling approach to model the relationship between time since infection and viral sequence diversity linearly using a baseline slope of all data from all individuals and gene-regions, in addition to individual-specific and gene-region-specific slope-modifying terms (Fig 1B). We represent observed time since infection differently depending on the timing of infection of each individual in the training data set. As described above, true infection timing was estimated using frequent HIV testing. However, there is uncertainty surrounding these HIV-testing-derived true time since infection measures due to limited testing frequency, especially for individuals who tested positive for HIV at birth and were infected in-utero. In an effort to reduce this uncertainty for in-utero-infected individuals, we chose to estimate "true" infection times directly within the modeling framework of the *infant-trained hierarchical model*. To do this, for individuals infected in-utero, we will use their age at sampling time as the "modeling time" for model fitting and estimate their infection time directly in the modeling framework. For individuals infected during or after birth, because there is more certainty in their HIV-testing-derived true times since infection, we will use these measures as the "modeling time" for model fitting.

We were motivated to use this approach since we are attempting to reduce uncertainty during model training while maximizing the timing information contained in these training data. However, because we do not have these model-derived "true" infection times when applying the *infant-trained hierarchical model* to held-out data, we use the HIV-testing-derived true time since infection measures for evaluating the prediction accuracy of the model, regardless of the timing of infection of each individual in the held-out data set. We describe the details of our model design and model validation below.

## Infant-trained hierarchical modeling notation

Let $I$ represent the set of all individuals in the training data set, $F$ represent the set of three gene regions used to calculate APD measures, and $J$ represent the set of all time point observations for each individual $i \in I$. We define observed time $t_{ij}$ to be the age of each individual $i \in I$ for observation $j \in J$. As such, $t_{ij} = 0$ represents birth and $-0.75 \leq t_{ij} < 0$ represents time in-utero (measured in years). We define infection time, $d_i$, to be the observed time (age) at which viral APD, $D_{if}$, is equal to 0 for each individual $i \in I$ and gene region $f \in F$. Thus, we define time since infection, $s_{ij}$, for each individual $i \in I$ and observation $j \in J$ such that

$$s_{ij} = t_{ij} - d_i.$$

As such, for each individual infected during delivery, time since infection will be equal to observed time ($s_{ij} = t_{ij}$ and $d_i = 0$). For each individual infected through breastfeeding (after birth), time since infection will be less than the observed time ($s_{ij} < t_{ij}$ and $d_i > 0$; Fig 8A). For

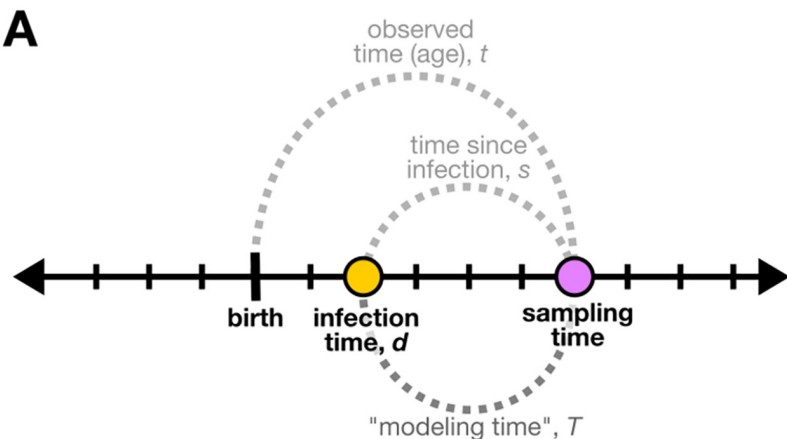

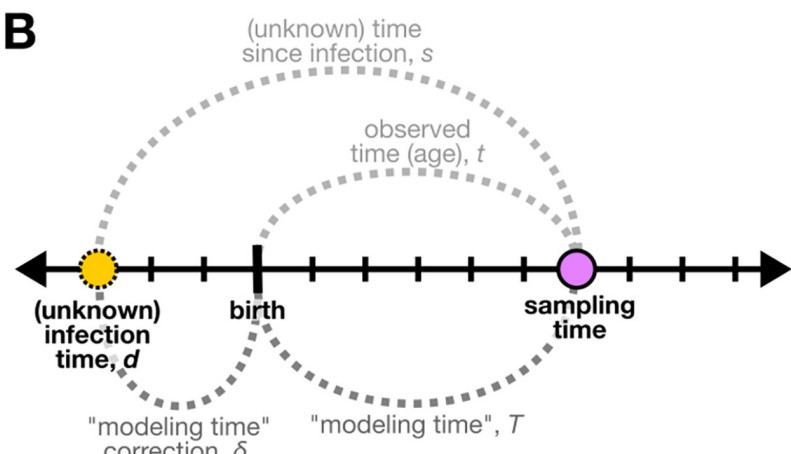

**Fig 8. Representation of "modeling time" $T$ in relation to time since infection $s$ and observed time $t$ for individuals infected after birth (A) and before birth (B). (A)** Individuals infected after birth have an infection time $d$ greater than zero. Given frequent HIV testing, we can closely estimate the true infection time for these individuals and, as such, closely estimate the true time since infection $s$ at a certain sampling time. In this example, the individual was infected at two months of age. If we sample their viral diversity at six months of age, then their time since infection at that time point is four months. For these individuals, we use this time since infection measure as the "modeling time" during our modeling. **(B)** Individuals infected in-utero have an infection time $d$ less than zero. As such, we cannot estimate the true infection time or the time since infection $s$ for these individuals using frequent HIV testing. In this example, the individual was infected three months before birth. If we sample their viral diversity at six months of age, then their time since infection at that time point (which we cannot estimate using HIV testing) is nine months. For these individuals, we use the age of the individual at sampling time, the observed time $t$, as the "modeling time" during our modeling. Because the observed time is not equivalent to time since infection (which is the value we want to estimate with our model), we infer a "modeling time"-correction term $\delta$ which is an estimate of the unknown infection time $d$ directly within the model framework.

each individual infected in-utero, time since infection will be greater than the observed time ($s_{ij} > t_{ij}$ and $d_i < 0$; Fig 8B).

## Defining infant-trained hierarchical model to predict time of infection

In order to define our *infant-trained hierarchical model* to estimate time since infection, $s_{ij}$, for each individual $i \in I$ and observation $j \in J$, we make the following assumptions about HIV

biology. We assume that infants born to mothers living with HIV can be infected in-utero ($-0.75 \leq d < 0$), during delivery ($d = 0$), or through breastfeeding ($d > 0$). For infants infected in-utero, time points within the third trimester of pregnancy ($-0.333 \leq d < 0$) are most likely to be the infection time [49,50]. Infants infected through breastfeeding are infected at some point after one month of age ($0.083 \leq d$). Further, we assume that viral APD, $D$, increases with time, but this rate may be different (and, perhaps even negative) for each individual and for each gene region.

With these assumptions, we have designed a Bayesian hierarchical regression model (which we refer to as the *infant-trained hierarchical model*) to predict the time since HIV infection using viral APD. Specifically, we model estimated time since infection $\widehat{s_{ifj}}$ for each individual $i$, gene region $f$, and observation $j$ using a linear function such that

$$\widehat{s_{ifj}} \sim Normal(m_{if} * D_{ifj} , \sigma_s^2)$$

$$\sigma_s^2 \sim Cauchy(0, 0.5)$$

where $\sigma_s^2$ is the estimated time since infection variance estimate, $m_{if}$ is the slope, and $D_{ifj}$ is the APD measure. We can define the slope $m_{if}$ for an individual $i$ and gene region $f$ to be

$$m_{if} = m + m_i + m_f$$

where $m$ is defined to be the baseline slope, $m_i$ is defined to be the individual-specific slope-modifying term, and $m_f$ is defined to be the gene-region-specific slope-modifying term. The baseline slope $m$ is modeled such that

$$m \sim Uniform(0, 150).$$

The individual-$i$-specific slope-modifying term $m_i$ is modeled such that

$$m_i \sim Normal(\mu_i, \sigma_i^2)$$

$$\mu_i \sim Normal(0, 1)$$

$$\sigma_i^2 \sim Cauchy(0, 20)$$

where $\mu_i$ is the individual-specific slope-modifying term mean estimate and $\sigma_i^2$ is the individual-specific slope-modifying term variance estimate. Likewise, the gene-region-$f$-specific slope-modifying term $m_f$ is modeled such that

$$m_f \sim Normal(\mu_f, \sigma_f^2)$$

$$\mu_f \sim Normal(0, 1)$$

$$\sigma_f^2 \sim Cauchy(0, 20)$$

where $\mu_f$ is the gene-region-specific slope-modifying term mean estimate and $\sigma_f^2$ is the gene-region-specific slope-modifying term variance estimate.

As described in the previous section, we will represent observed time since infection differently depending on the infection time estimate of each individual in the training data set, and we will define the variable "modeling time" to contain these observed time since infection measures. For individuals infected during or after birth (with infection time, $d \geq 0$), true infection time and times since infection were estimated using frequent HIV testing, and we will use

these observed times since infection as the "modeling time" for model fitting (Fig 8A). However, for individuals infected in-utero (with infection time, $d < 0$), because their true infection time was unmeasurable, we will instead use their age at sampling time as the "modeling time" for model fitting and estimate their infection time directly in the modeling framework using a "modeling time"-correction term which we will describe later (Fig 8B).

Specifically, we define modeling time $T_{ifj}$ for each individual $i$, gene region $f$, and observation $j$ such that $T_{ifj} = s_{ij}$ for individuals infected after birth and $T_{ifj} = t_{ij}$ for individuals infected before birth (in-utero), where $s_{ij}$ represents the time since infection estimates from frequent HIV testing and $t_{ij}$ represents the observed time (age) (Fig 8). We can model this observed modeling time $T_{ifj}$ such that

$$T_{ifj} \sim Normal(\widehat{T_{ifj}}, \ 0.1)$$

where $\widehat{T_{ifj}}$ represents the estimated modeling time (from our model as described below) and 0.1 represents the modeling time variance. As described previously, each individual was tested for HIV at 1–3 month intervals and the true infection time was predicted to be the midpoint between the last negative test and the first positive test (see the "Estimating true HIV infection time" section for further details). As such, this modeling time variance value aims to represent the possible error intrinsic to the way in which true infection time was predicted during data collection.

For each individual $i \in I$ infected in-utero ($d_i < 0$), since the infection time $d_i$ is not observed and since we are using observed time (age) $t_{ij}$ as the observed modeling time $T_{ifj}$ for model fitting, we define a "modeling time"-correction term $\delta_i$ that we will model directly as an estimate for $|d_i|$ (Fig 8B). Because time points within the third trimester of pregnancy ($-0.333 \leq d < 0$) are more likely to be the infection time [49,50], "modeling time"-correction values between 0 and 0.333 should be the most likely. As such, we model $\delta_i$ such that

$$\delta_i \sim Beta(1, \ 5).$$

When estimating the posterior distribution of these "modeling time"-correction parameters for in-utero-infected individuals using Hamiltonian Markov Chain Monte Carlo sampling, we found that the posterior means ranged from 0.96–2.28 months and the 89% credible intervals were quite narrow (i.e. the widest 89% credible interval was [0.24, 5.40] months). For each individual $i \in I$ infected during delivery or through breastfeeding ($d_i \geq 0$), since the infection time $d_i$ was directly estimated using frequent HIV testing and since we are using time since infection estimates $s_{ij}$ as the observed modeling time $T_{ifj}$ for model fitting, the modeling-time-correction term $\delta_i$ is not required and will equal 0 (Fig 8A).

With these "modeling time"-correction term estimates ($\delta_i$), we model the estimated modeling time, $\widehat{T_{ifj}}$, for each individual $i$, gene region $f$, and observation $j$ as

$$\widehat{T_{ifj}} = \widehat{s_{ifj}} \ - \ \delta_i$$

where $\widehat{s_{ifj}}$ is the estimated time since infection which we model as described above.

We used Hamiltonian Markov Chain Monte Carlo sampling to estimate a full posterior distribution over model parameters (Fig 9), given the infant training data set described above using the `rstan` [51] package in R.

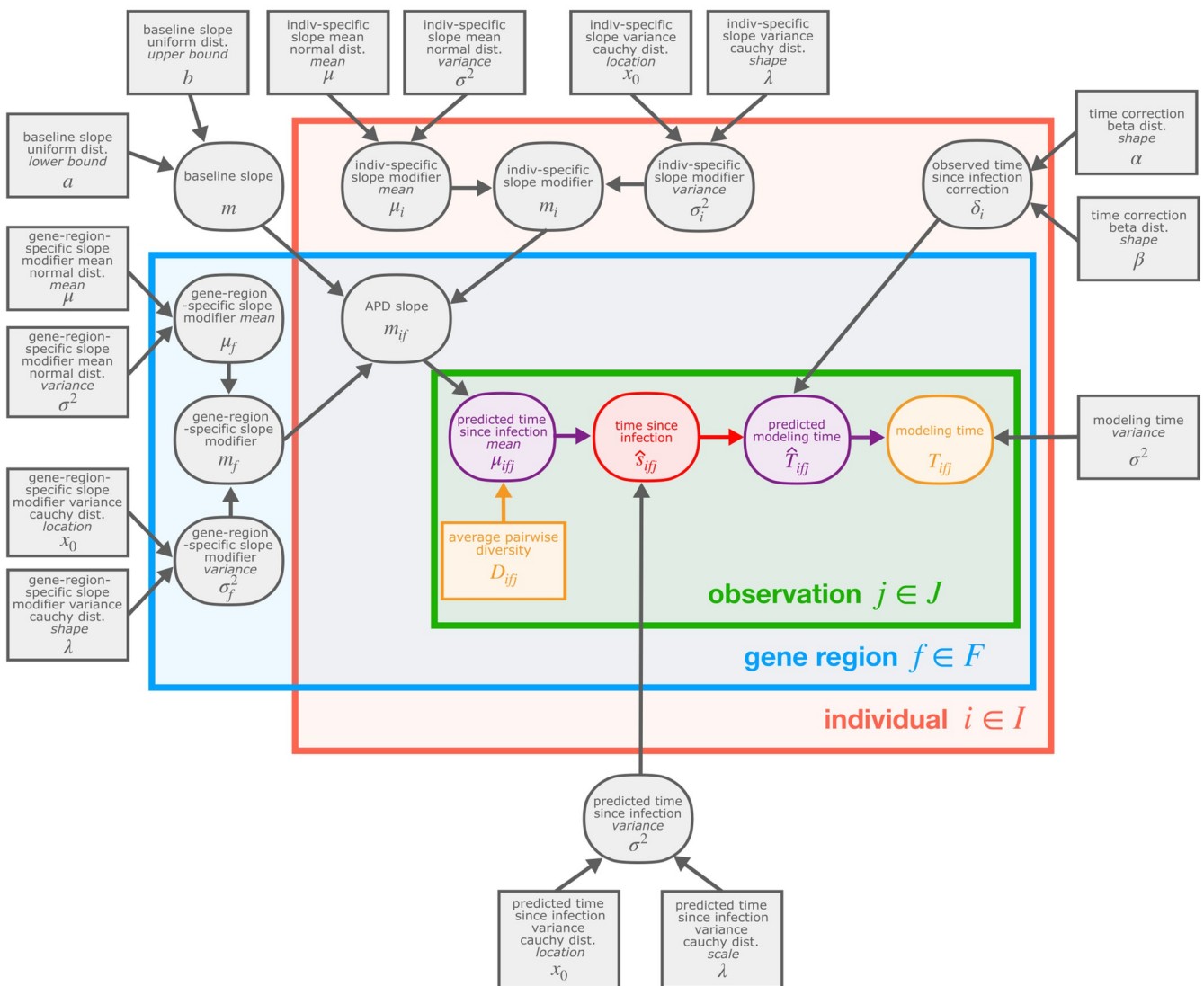

**Fig 9. Many infant-trained Bayesian hierarchical model parameters are estimated for each observation *j* ∈ *J*, gene region *f* ∈ *F*, and individual *i* ∈ *I* in the training data set when fitting the model.** In this plate diagram, rounded nodes represent parameters that are estimated during model fitting and square nodes represent constant values. The orange nodes represent observed data, the purple nodes represent deterministic values, and the red node represent the value, time since infection, that we are ultimately interested in estimating using our model.

## Alternate version of the infant-trained hierarchical model using Laplace distribution

We noted that residual errors from least absolute deviation regression (which was used to train the *infant-trained linear models* and the *adult-trained linear models*) are assumed to be randomly distributed with a Laplace distribution whereas errors from the *infant-trained hierarchical model* were designed to be normally distributed. Because the Laplace distribution features longer tails compared to the Normal distribution, regression methods involving Laplace distributions typically provide robust solutions even when outliers are present in the data. To explore whether this distinction affected the relative prediction accuracy of each model, we designed a version of the *infant-trained hierarchical model* in which observations were drawn from a Laplace distribution.

### Defining infant-trained linear models to predict time of infection

Adult studies have shown that rates of viral diversification can be used to accurately estimate infection timing using a least absolute deviation linear regression model [1]. As such, in addition to the *infant-trained hierarchical model*, we also trained least absolute deviation linear regression models using the infant training data set described above. Like the previously-published *adult-trained linear models*, we used least absolute deviation regression to train an infant model for each gene region separately. We refer to these models as the *infant-trained linear models*.

### Model validation

For each infant-trained model (e.g. the *infant-trained linear models* and the *infant-trained hierarchical model)*, we used leave-one-out-cross-validation to estimate time since infection for each individual and APD measure within the infant training data set. To do this, we removed all data for each individual from the data set (as opposed to data from one sampled time point for each individual), re-fit the model, and estimated time since infection for the held-out individual. We repeated this procedure for all individuals in the training data set. We measured prediction accuracy by comparing these model-derived time since infection estimates to the HIV-testing-derived true time since infection measures for each individual. We compared the accuracy of these predictions to those from the *adult-trained linear models*.

Further, we defined an independent testing data set containing single-time-point viral APD measures from 15 additional infants (described above). Data from these infants was not present in the original training data set. Before analyzing these data, we "froze" our trained model coefficients (for the *infant-trained linear models*) and posterior distributions over model parameters (for the *infant-trained hierarchical model*) in git commit 43f312c on our repository. We then inferred the time since infection for each of the 15 testing data set infants using these frozen regression coefficients for each infant-trained model (i.e. *infant-trained hierarchical model* and *infant-trained linear models)*. As before, we compared the accuracy of these predictions to those from the *adult-trained linear models*.

## Supporting information

**S1 Fig. APD increases with time for most individuals and gene regions.** The inferred distribution of APD slope for each individual and gene region in the training data set obtained using Markov Chain Monte Carlo sampling from the Bayesian hierarchical model (described above) is shown. The median slope values are plotted, along with the 89% credible interval for the parameter distribution. An APD slope equal to zero is shown by the horizontal dashed line. APD slopes are shown in units months/diversity–as such, a higher rate indicates that diversity is accumulating slower with time.
(TIF)

**S2 Fig. APD slope does not vary substantially by rate of CD4+ T cell decline.** The inferred median APD slope for each individual and gene region from the *infant-trained hierarchical model* is shown as a function of the rate of CD4+ T cell percentage decline for each individual.
(TIF)

**S3 Fig. Posterior predictive checks show that the replicated data generated under the *infant-trained hierarchical model* look similar to the observed data in terms of median and standard deviation, but not in terms of minimum or maximum.** The (**A**) median, (**B**) standard deviation, (**C**) maximum, and (**D**) maximum test statistics for the distribution of true observed time since infection measures are shown by the vertical black line. The p-values from

comparing each test statistic computed for the true observed time since infection measures to the observed time since infection measures simulated from the model are also shown. The unit for each test statistic is months.
(TIF)

**S4 Fig. Comparison of the model prediction errors for the training data set when drawing observations from a Laplace distribution and a Normal distribution within the infant-trained hierarchical model formulation.** Leave-one-out-cross-validation was used to calculate the estimated time since infection for each individual. (A) Drawing observations from a Laplace distribution instead did not improve prediction accuracy. The difference between true time since infection and model-derived time since infection is shown on the x-axis. A difference of zero is shown by the black vertical line. (B) Comparison of true time since infection and model-derived time since infection for both models using APD measures sampled from gene region 3. The 89% credible interval is shown as a vertical black line for each estimate. The blue line represents the model-derived time since infection = true time since infection line. The gray line shows the line of best fit.
(TIF)

**S5 Fig. Distribution of APD measures for the three gene-regions for each data set.** The testing data set has much higher APD measures within gene-region 1 compared to the training data set.
(TIF)

**S6 Fig. Distribution of time since infection measures for each data set.** The testing data set has much higher time measures compared to the training data set.
(TIF)

**S1 Table. Description of Bayes factor cutoffs and interpretations.**
(PDF)

**S2 Table. HIV-1 amplification primers.**
(PDF)

## Acknowledgments

We thank the participants and staff of the Cytotoxic T Lymphocyte cohort and the Nairobi Breastfeeding Trial. We also thank Hassan Nasif, Thayer Fisher, Will Dumm, Chris Jennings-Shaffer, and Jiansi Gao for helpful discussions regarding this paper, as well as Morgane Rolland for sharing example BEAST XML files from [43].

This article is subject to HHMI's Open Access to Publications policy. HHMI lab heads have previously granted a nonexclusive CC BY 4.0 license to the public and a sublicensable license to HHMI in their research articles. Pursuant to those licenses, the author-accepted manuscript of this article can be made freely available under a CC BY 4.0 license immediately upon publication.

## Author Contributions

**Conceptualization:** Julie Overbaugh, Grace John-Stewart, Dara A. Lehman.

**Data curation:** Carolyn S. Fish, Sara Drescher, Noah A. J. Cassidy, Pritha Chanana, Jennifer Slyker, Barbra Richardson, Dara A. Lehman.

**Formal analysis:** Magdalena L. Russell, Carolyn S. Fish, Sara Drescher, Noah A. J. Cassidy, Pritha Chanana, Barbra Richardson.

**Funding acquisition:** Julie Overbaugh, Grace John-Stewart, Frederick A. Matsen, IV, Dara A. Lehman.

**Investigation:** Magdalena L. Russell, Carolyn S. Fish, Sara Drescher, Noah A. J. Cassidy.

**Methodology:** Magdalena L. Russell, Carolyn S. Fish, Sara Drescher, Noah A. J. Cassidy, Pritha Chanana, Grace John-Stewart, Frederick A. Matsen, IV, Dara A. Lehman.

**Project administration:** Sarah Benki-Nugent, Jennifer Slyker, Dorothy Mbori-Ngacha, Rose Bosire, Dalton Wamalwa, Elizabeth Maleche-Obimbo, Julie Overbaugh, Grace John-Stewart, Dara A. Lehman.

**Resources:** Jennifer Slyker, Dorothy Mbori-Ngacha, Rose Bosire, Dalton Wamalwa, Elizabeth Maleche-Obimbo, Julie Overbaugh, Grace John-Stewart, Dara A. Lehman.

**Software:** Magdalena L. Russell.

**Supervision:** Dorothy Mbori-Ngacha, Rose Bosire, Dalton Wamalwa, Elizabeth Maleche-Obimbo, Julie Overbaugh, Grace John-Stewart, Frederick A. Matsen, IV, Dara A. Lehman.

**Validation:** Magdalena L. Russell, Carolyn S. Fish, Sara Drescher, Noah A. J. Cassidy, Pritha Chanana, Barbra Richardson.

**Visualization:** Magdalena L. Russell.

**Writing – original draft:** Magdalena L. Russell, Dara A. Lehman.

**Writing – review & editing:** Magdalena L. Russell, Carolyn S. Fish, Sara Drescher, Noah A. J. Cassidy, Sarah Benki-Nugent, Jennifer Slyker, Barbra Richardson, Dalton Wamalwa, Elizabeth Maleche-Obimbo, Julie Overbaugh, Grace John-Stewart, Frederick A. Matsen, IV, Dara A. Lehman.

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
