## [Decision Letter · Decision Letter 0]

1 Oct 2023

Dear Dr. Lehman,

Thank you very much for submitting your manuscript "Using viral sequence diversity to estimate time of HIV infection in infants" for consideration at PLOS Pathogens. As with all papers reviewed by the journal, your manuscript was reviewed by members of the editorial board and by several independent reviewers. In light of the reviews (below this email), we would like to invite the resubmission of a significantly-revised version that takes into account the reviewers' comments.

We cannot make any decision about publication until we have seen the revised manuscript and your response to the reviewers' comments. Your revised manuscript is also likely to be sent to reviewers for further evaluation.

Sincerely,

Daniel C. Douek

Academic Editor

PLOS Pathogens

Richard Koup

Section Editor

PLOS Pathogens

Kasturi Haldar

Editor-in-Chief

PLOS Pathogens

orcid.org/0000-0001-5065-158X

Michael Malim

Editor-in-Chief

PLOS Pathogens

orcid.org/0000-0002-7699-2064

Reviewer's Responses to Questions

**Part I - Summary**

Reviewer #1: Russell and co-authors introduce a Bayesian hierarchical model for estimating timing of infection in infants. They show that training their model on the diversity of viral sequences sampled in utero and postpartum more accurately predicts infection timing than training on sequences sampled from adults (or linear modeling trained on infant sequences). Using this method, they find considerable inter-individual and gene region variation, but overall that pol diversifies faster in infants than adults, irrespective of SPVL or CD4+ decline. The authors have done well to develop a method that takes advantage of their unique dataset and the results are promising for its further use. I have a few outstanding questions regarding how the method was tested and how it will be more widely implemented. -Eric Lewitus

Reviewer #2: In their manuscript, ‘Using viral sequence diversity to estimate time of HIV infection in infants’, estimated rates of viral diversification in 22 Kenyan children and determined that this rate was three times higher than what is seen in infected adults. In this population the diversification rate did vary by individual, route of infection (during delivery vs. breast feeding) and by viral gene region analyzed. However, set point viral load or rate of CD4 decline did not seem to be associated with differences in viral diversification rates. The authors used a Bayesian hierarchical model trained on either the infant data or existing adult data to determine the best fit to the data from the validation cohort of 15 different infants. They found that the model trained with the data from the Kenyan infants fit the validation model best. Unfortunately, there is too much subject-to-subject variability to differentiate infants who were infected in-utero vs. after birth. Overall, this is a well written and thorough investigation. The figures and tables are clear and appropriate. There are some issues to address.

**Part II – Major Issues: Key Experiments Required for Acceptance**

Reviewer #1: The state-of-the-art for inferring timing of infection is phylogenetic molecular dating. I think the utility of the authors’ method would need to be shown to be superior in ease of implementation and at leats comparable in results to, for example, inferences of timing of infection drawn from BEAST analyses. A comparison to Dearlove et al., PLoS Comp Bio, 2021 (or another relevant analysis) would suffice. If Bayesian phylogenetic analysis for molecular dating is not appropriate for their question (or infant sequences), then it would be helpful for the authors to explain why.

Timing of infection is inferred in years. Given that the individuals are infants, which gives a small window for timing of infection, and that MAE is around 0.5 years, is this enough resolution? What questions regarding timing of infection do the authors imagine would be answered at this resolution? As results are given in fractions of years (e.g., 1.68), it would be useful to report results in months. Otherwise, please explain why years were chosen as units.

Reviewer #2: (No Response)

**Part III – Minor Issues: Editorial and Data Presentation Modifications**

Reviewer #1: Line 145: The studies cited supporting that “the majority of infants are infected with only a single viral variant” are quite limited (3-10 mother-infant pairs) and in one study (Wu et al., 2006) 2/8 infant infections appear to have been established by multiple founder variants, which is on par with adults. As deep sequencing is showing that rare founder variants are more common than previously thought, how would this affect inference with the Bayesian hierarchical model. Can it be adapted to infer timing in multi-founder infections? If not, I think it is worth saying that the method is only application to infections established with a single variant.

Line 157: Considering that certain sites will always be conserved, wouldn’t it make more sense to compute the APD backwards in time to when diversity was zero? A results of 105 years — which if I understand correctly means that it would take 105 years for all sites between two sequences to be mismatched — is difficult to conceptualize. Perhaps the authors could explain further why they chose to look at the rate of diversity accumulation forwards in time.

Lines 225-237: If the authors “did not use this alternative model for downstream analyses”, why was it presented? Could the authors explain what this alternative model brings to the manuscript?

Lines 427-429: If individuals 12 and 13 did not test positive at birth, why would they have elevated rates compared to individuals infected in utero and test positive at birth? Does the difference between in utero and postpartum rates disappear if these individuals are removed?

Lines 466-477: Could the model instead be fit with a non-linear model? Or something responsive to Ne?

Line 481: Were there any differences in mothers who received ART during pregancy?

Reviewer #2: 1) The last sentence of the introduction, on line 61, the authors state that, “These findings hold promise for developing infant-specific treatment approaches and preventive measures.” What are the possible treatment procedures and prevention measures that would use the findings presented in the manuscript? This should be addressed in in the discussion.

2) On line 481, the authors state that some of the mothers were on ART at the time of birth. Why was exposure not added to the model to see if it mattered?

3) Line 425 states, “In fact, we found that only one individual (individual 485) had a decrease in viral diversity over time across all gene regions.” However, only individuals numbered 1 to 21 are presented. Based on Figure 1 panel A, I believe the authors are referring to individual #7.

4) It is surprising that the rates of diversification for HIV are three-fold faster in infants than in adults. This is especially true given the infants undeveloped immune system at the time of infection. In adults whose immune systems are severely compromised (like end stage AIDS), the rates of diversification are less than when their immune system is intact (see Shankarappa J Virol 1999 for example). These individuals also have very high viral loads, similar to that seen in infants. Adaptation to a new a new host could be the reason for the increased viral evolution that is observed. This would be supported by the author’s observation that the rate of diversification decreases after the first year.

5) How would the impact of accelerated HIV sequence evolution in infants impact other studies? For example, the outbreak in children attending the Al-Fateh Hospital in Benghazi, Libya (see de Oliveira Nature 2006), if a different rate of diversification for the infants was used would that implicate the foreign medical staff in infecting the infants?

PLOS authors have the option to publish the peer review history of their article (what does this mean?). If published, this will include your full peer review and any attached files.

Reviewer #1: **Yes: **Eric Lewitus

Reviewer #2: No
---

## [Decision Letter · Decision Letter 1]

27 Nov 2023

Dear Dr. Lehman,

We are pleased to inform you that your manuscript 'Using viral sequence diversity to estimate time of HIV infection in infants' has been provisionally accepted for publication in PLOS Pathogens.

Best regards,

Daniel C. Douek

Academic Editor

PLOS Pathogens

Richard Koup

Section Editor

PLOS Pathogens

Kasturi Haldar

Editor-in-Chief

PLOS Pathogens

orcid.org/0000-0001-5065-158X

Michael Malim

Editor-in-Chief

PLOS Pathogens

orcid.org/0000-0002-7699-2064

Reviewer Comments (if any, and for reference):

Reviewer's Responses to Questions

**Part I - Summary**

Reviewer #2: (No Response)

**Part II – Major Issues: Key Experiments Required for Acceptance**

Reviewer #2: (No Response)

**Part III – Minor Issues: Editorial and Data Presentation Modifications**

Reviewer #2: (No Response)

PLOS authors have the option to publish the peer review history of their article (what does this mean?). If published, this will include your full peer review and any attached files.

Reviewer #2: **Yes: **Oliver Laeyendecker

---

## [Editor Report · Acceptance letter]

4 Dec 2023

Dear Dr. Lehman,

We are delighted to inform you that your manuscript, "Using viral sequence diversity to estimate time of HIV infection in infants," has been formally accepted for publication in PLOS Pathogens.

Best regards,

Michael Malim

Editor-in-Chief

PLOS Pathogens

orcid.org/0000-0002-7699-2064